# Preparation of Composite Monolith Supercapacitor Electrode Made from Textile-Grade Polyacrylonitrile Fibers and Phenolic Resin

**DOI:** 10.3390/ma13030655

**Published:** 2020-02-01

**Authors:** Karim Nabil, Nabil Abdelmonem, Masanobu Nogami, Ibrahim Ismail

**Affiliations:** 1Zewail City of Science and Technology, Giza 12578, Egypt; imohamed@zewailcity.edu.eg; 2Chemical Engineering Department, Cairo University, Giza 12613, Egypt; nabil_abdelmonem@yahoo.com; 3Department of Electric and Electronic Engineering, Kindai University, Osaka 577-8502, Japan; mnogami@ele.kindai.ac.jp

**Keywords:** textile-grade polyacrylonitrile fibers, phenolic resin, composite monolith, thermal stabilization of PAN fibers, electrochemical double-layer supercapacitor, EDLC

## Abstract

In this work a composite monolith was prepared from widely available and cost effective raw materials, textile-grade polyacrylonitrile (PAN) fibers and phenolic resin. Two activation procedures (physical and chemical) were used to increase the surface area of the produced carbon electrode. Characterization of the thermally stabilized fibers produced was made using differential scanning calorimetry (DSC), thermal gravimetric analysis (TGA) and Carbon-Hydrogen-Nitrogen(CHN) elemental analysis, in order to choose the optimum conditions of producing the stabilized fibers. Characterization of the produced composite monolith electrode was performed using physical adsorption of nitrogen at 77 °K, cyclic voltammetry (CV), galvanostatic charge-discharge (GCD) and electrical resistivity in order to evaluate its performance. All the electrodes prepared had a mixture of micropores and mesopores. Pressing the green monolith during the curing process was found to reduce largely the specific surface area and to some degree the electrical resistivity of the chemically activated composite electrode. Physical activation was more suitable than chemical activation, where it resulted in an electrode with specific capacity 29 F/g, good capacitive behavior and the stability of the electrical resistivity over the temperature range −130 to 80 °C. Chemical activation resulted in a very poor electrode with resistive rather than capacitive properties.

## 1. Introduction

Recently, there is a rapid market demand for efficient energy in technological fields such as hybrid electric vehicles, back-up power supplies, portable electronic devices, adjustable speed drives, off-peak energy storage, improvement in power quality and high power actuators [1].

Electrochemical energy storage devices such as electrochemical supercapacitors, fuel cells and batteries are considered promising energy sources [2]. Electrochemical supercapacitors can be classified into three classes based on the mechanism of electric charge storage, electrochemical double layer capacitors (EDLC), pseudocapacitors and hybrid capacitors [3]. Particularly, EDLC are the most favorable energy storage and conversion devices, due to some advantages such as: 1—Storing more energy than conventional capacitors and deliver more power than electric batteries. 2—Very fast charging and discharging. 3—Long cycle-life exceeding 100,000 cycle [1]. 

An electrochemical supercapacitor consists of two electrodes, an electrolyte, and a separator [3]. There are four types of electrolytes: aqueous electrolytes (e.g., H_2_SO_4_ and KOH), organic electrolytes (e.g., acetonitrile and propylene carbonate), ionic liquids (e.g., liquid salts) [4], solid state electrolytes (e.g., polyvinyl alcohol mixed with potassium ferrocyanide and potassium ferricyanide gel) [5,6].

Carbonaceous nano-materials such as graphene, carbon nanotubes and activated carbons are the most utilized electrode materials for EDLC due to their efficient characteristics as high surface area, electrochemical stability, high conductivity, and environmental friendliness. [7]. However, research efforts to utilize graphene as a supercapacitor electrode are still in progress to prevent stacking of graphene layers which limits and reduces the specific capacity of the produced supercapacitor electrode. On the other hand, carbon nanotubes have large stacking problems too, that make their manufacturers decrease their production in the last years [8]. The production of both graphene and carbon nanotubes is limited due to the complexity and relatively high cost of manufacturing [2].

Activated carbons made of biomasses have a large porosity and defective structure which results in low conductivity and low electrochemical stability, while their randomly connected pores reduce ionic transport. Due to these factors, activated carbons rarely display high power density [8].

There are two other forms of carbon—carbide-derived carbon and templated carbon—which were both developed as supercapacitor electrodes. The manufacturing processes of these two types of carbon have a low carbon yield, high cost, and environmental concerns. Thus, their commercial potential is limited.

The cost of manufacturing carbon and thereafter its processing into supercapacitor electrode is a crucial factor in choosing the optimum type of carbon material to be used as supercapacitor electrode because the market needs a reduction in the price of supercapacitors in order to be widespread and have commercial appeal [8].

Textile-grade polyacrylonitrile fibers can be utilized in the manufacture of carbon fibers [9], but the produced carbon fibers are of lower performance (i.e., lower strength and stiffness). These lower performance carbon fibers may be used as reinforcement of plastic materials, electrically conducting fillers for polymers, and when activated they are useful in adsorption applications [10]. These properties of PAN fibers qualify them to be employed in preparation of activated carbon fiber composite monolith which can be used finally as an electrode for EDLC.

Phenolic resins are characterized by their low cost and chemical resistance besides their increased electrical conductivity with increasing carbonization temperature [11]. Because of these attractive properties, it is encouraging to use phenolic resin as a binding material in the fabrication of an EDLC composite electrode. 

Given the aforementioned considerations regarding different forms of carbon (graphene, carbon nanotubes, activated carbon, carbide-derived carbon, and templated carbon), we aim in this work to fabricate an EDLC electrode in the form of a composite monolith from readily available and relatively inexpensive substances found in the market and to employ simple steps in the fabrication process in order to reduce the cost of EDLC fabrication in the market.

## 2. Materials and Methods 

Materials used in this study are as follows:

As-received fibers: Textile-grade 100% Acrylic fibers (supplier: Fisipe Manufacturer, Belgium);

Borax: commercial grade sodium tetraborate decahydrate (Na_2_B_4_O_7_.10H_2_O) (supplier: Siag Chemicals Group, Egypt);

Phenolic resin: cold-setting resole resin (supplier: Modern for Chemical Industries Company, Egypt);

Potassium hydroxide: laboratory grade (supplier: ELNASR for Chemicals and Drugs, Egypt).

### 2.1. Procedure of Preparing the Carbon Composite Green Monolith

The schematic for the preparation of the composite green monolith is depicted in Figure 1.

### 2.2. Procedure (1): Chemical Activation of the Composite Green Monolith

Subsequently, there were two different procedures of carbonization and activation of the prepared green monolith which are depicted in Figure 2 and Figure 3.

### 2.3. Procedure (2): Physical Activation of the Composite Green Monolith

In procedure (1), the prepared carbon monolith electrodes corresponding to the three different concentrations (2, 3, and 5 molar) are termed as sample (A), sample (B) and sample (C), respectively.

In procedure (2), the prepared carbon monolith electrodes corresponding to the three different carbonization temperatures (700, 800, and 900 °C) are termed as sample (D), sample (E) and sample (F), respectively.

### 2.4. Characterization of Stabilized Fibers 

Three tests were performed in order to characterize the fibers after the thermal stabilization process, also to optimize the conditions of this processing step. These tests are differential scanning calorimeter (DSC, TA Instruments, New Castle, DE, USA), thermal gravimetric analysis (TGA, TA Instruments, New Castle, DE, USA), and elemental analysis (Elementar Analysen Systeme GmbH, Langenselbold, Germany).

Differential Scanning Calorimetery: SDT Q600 calorimeter (TA Instruments, New Castle, DE, USA) was employed to determine the temperature at which the precursor fibers undergo exothermic reactions, and record the changes in the nature of the exotherms and their extent. The raw fibers were tested under air environment and heated to 400 °C, while the stabilized fibers were tested under nitrogen gas and heated up to 1000 °C.

Thermal Gravimetric Analysis: SDT Q600 calorimeter was also employed to measure the weight loss of fibers from room temperature up to 1000 °C under N_2_ inert gas, and to determine which of the thermally stabilized fibers had the greatest yield after the stabilization process.

Elemental Analysis: Vario EL ||| Germany equipment was employed to determine the mass ratios of carbon, nitrogen, and hydrogen elements with respect to the mass of the stabilized fibers.

### 2.5. Characterization of Carbonized and Activated Composite Monolith Electrode 

Three analytic tests were performed to characterize and evaluate the performance of the produced composite monolith electrode after being carbonized and activated. These are surface area and pore volume distribution test, electrochemical test, and electrical resistivity test.

#### 2.5.1. Surface Area and Pore Volume Distribution Characterization 

This test was performed to measure the specific surface area, pore volume, and the pore size distribution of the prepared electrodes using NOVAtouch ™ Equipment, version 1.2 (Quantachrome Instruments, Graz, Austria). The method of investigation was N_2_ adsorption-desorption isotherm measurement at 77 °K. 

#### 2.5.2. Electrochemical Characterization:

Electrical Impedance Spectroscopy (EIS), Cyclic Voltammetry (CV) and Galvanostatic Charge–Discharge (GCD) analyses were performed using Autolab PGSTAT302N instrument (Metrohm Autolab, Utrecht, The Netherlands) in order to determine the specific capacity of the prepared supercapacitor electrode and to evaluate the electrochemical performance. In the CV test, three-electrode configuration was used, where a linearly changed potential between the working electrode and the reference electrode was applied. Different scan rates (1, 10, 25, 50 and 100 mv/sec) were used for all samples in the CV test. The voltage range used for all samples in EIS test was (0 to 1 V), while that used in CV and GCD tests was (−0.5 to 1 V). The frequency range employed in EIS test was (0.01 Hz–100 KHz). The electrolyte was a 2 Molar H_2_SO_4_ solution.

#### 2.5.3. Electrical Resistivity Measurement:

The electrical resistivity of the prepared electrodes was measured using Lucas/Signatone Four Point Probe SP4-HT4 instrument (Lucas/Signatone Corporation, California, USA) to assess the electrical conductivity of the prepared composite monolith samples. Thin film method was applied.

## 3. Results and Discussion

### 3.1. Differential Scanning Calorimetry 

Table 1 shows the amount of the energy released in exothermic reactions occurred within the as-received fibers. It also shows the temperature at the beginning of the exothermic peak and the temperature at the vertex of the peak. These exothermic reactions are cyclization reactions [12]. Figure 4 shows the DSC Thermograms of the as-received fibers and the stabilized fibers in different treatment time periods 0.5, 1, 2, and 3 h.

DSC analysis of the as-received fibers and thermally stabilized fibers revealed that only the as-received fibers have an exothermic peak of only 251.3 J/g starting at temperature 305.9 °C, while the fibers resided for different periods of time within the furnace during stabilization have no such peak. The elimination of the exothermic peak means that the fibers were completely stabilized [13]. This elimination of the exothermic peak is what concerns us here, because these exothermic reactions are the reason for the problems of hardening, coalescence and breakage of fibers, therefore one can choose any of the four time periods (0.5, 1, 2, 3 h) for the residence of the fibers in the furnace. The optimum residence time will be the least one in order to reduce the processing time and the manufacturing cost. 

### 3.2. TGA of Thermally Stabilized Fibers

Figure 5 shows the TGA analysis of the as-received fibers and the four samples of fibers resided in the furnace for (0.5, 1, 2, 3 h), and Table 2 displays their residual weight percent at 990 °C. An important information which can be obtained from this analysis, is that the residual weight percent at the highest analyzing temperature, here it is 990 °C, represents the yield of the thermal stabilization process. 

TGA analysis of the as-received fibers shows that the residual weight is 30.125 % only of the original weight of the fibers. TGA analysis of the thermally stabilized fibers revealed that the fibers left for (0.5, 1, 2, and 3 h) at 260 °C in the stabilization furnace lost (56.4 %, 48.9 %, 48.57 %, and 58.39 %) of their initial weights, respectively. What concerns us here is to discover the maximum weight percent remained at the final analysis temperature 990 °C, which corresponds to the maximum yield of the carbonized fibers. It is clear from Table 2 that soaking the fibers for 2 h in the furnace at 260 °C gives the maximum yield (51.4%).

### 3.3. Elemental Analysis of Thermally Stabilized Fibers

Table 3 shows the carbon, hydrogen, and nitrogen weight percents with respect to the weight of the as-received fibers and the thermally stabilized fibers. It also shows the ratios of carbon element to both hydrogen element and nitrogen element.

The important needed information from this analysis was to observe the percent of carbon to other elements, since carbon is the electrical conducting element and its increase at the expense of other non-conducting elements will enhance the electrical conductivity of the produced electrode. The elemental analysis shows that the carbon percent in the fibers first decreases gradually with rise in treatment time from 0.5 to 2 hours and then increases suddenly at 3 hours Getting both the ratios of carbon element to nitrogen element, and of carbon element to hydrogen element, they were found at their highest at the residence time of 2 hours These results suggest that the resided time (2 hours) is the optimum since what is finally needed is a fiber with carbon content which is higher than other elements.

Based on the above three tests of DSC, TGA, and elemental analysis, it is obvious that the residence time of 2 hours within the furnace represents the optimum time, this residence time will be applied here in this work for further treatment procedures. 

### 3.4. Results of Obtaining the Carbonized and Activated Composite Monoliths

Results of chemically activated samples (procedure 1) and those of physically activated samples (procedure 2) are presented in the following sections. 

#### 3.4.1. Porous Structure Properties

Carbon EDLC electrodes have pores embedded within the carbon particles (intra-particle porosity), these pores are fine nanopores with sizes (< 5 nm), also they have larger nano- and micro-pores (inter-particle porosity) created by the packing of carbon particles. A large part of the electrode surface area and hence the majority of the capacity is associated within the intra-particle pores. On the other hand, the percolation and accessibility of electrolyte ions to the fine pores is affected by the network of inter-particle pores which act as electrolyte pathways. If the electrode structure is closely packed, the amount of the electrolyte ions resided within the inter-particle pores (formed between irregular carbon particles) will be reduced, leading to electrolyte depletion and minimum capacity [14]. 

The porosity of samples (A), (B) and (C) that were activated chemically and samples (D), (E) and (F) that were activated physically was investigated by using N_2_-adsorption isotherms (Figure 6 and Figure 7). The size distribution of micropores (pores with diameter < 2 nm) was obtained by the Dubinin–Astakhov (DA) method (Figure 8 and Figure 9) [15], while the size distribution of mesopores (pores with diameter > 2 nm and < 50 nm) was obtained by the Barrett–Joyner–Halenda (BJH) method (Figure 10 and Figure 11) [16].

According to the International Union of Pure and Applied Chemistry (IUPAC) data classification, the isotherm profiles of samples (A, B and C) are compatible with type Ⅳ pattern, this pattern characterizes adsorption on mesoporous solids. The isotherm profiles of samples (D, E and F) are compatible with type І pattern, this pattern represents adsorption by microporous solids [17,18].

Table 4 shows the calculated values of the specific surface area S_BET_ of all the samples (A, B, C, D, E, and F) based on multipoint Brunauer–Emmett–Teller (BET) method, besides the specific surface area of micropores (S_micro_) and mesopores (S_meso_). It also shows the specific volume of micropores (V_micro_) and mesopores (V_meso_).

The results in Table 4 show that, for the chemically activated samples (A, B and C), there is a trend of decreasing of the specific surface area by increasing the concentration of the KOH solution. Both the mesopores volume and mesopores surface area for each sample are slightly larger than those of micropores, therefore these samples have more mesopores content than the micropore content. For the physically activated samples (D, E and F), sample (D) has the largest specific surface area S_BET_ = 486 m^2^/g, while samples (E) and (F) have lower surface area S_BET_ = 433 m^2^/g and 405 m^2^/g, respectively. This indicates that carbonization at lower temperatures (700 °C) is beneficial to the material with respect to increasing its surface area, while using higher carbonization temperatures (800 °C and 900 °C) leads to lowering the specific surface area of the material. For samples (D, E and F), the situation is different from samples (A, B and C), where the mesopores volume and mesopore’s surface area for each sample is significantly smaller than those of micropores, thus these samples have more micropores content than the mesopore content. As shown in Table 4, samples (D, E and F) had more specific surface area than samples (A, B and C), this may be partly due to the pressing of the green monolith for samples (A, B and C) during the molding process, where this pressing process helps in compacting the monolith and blocking the voids. This compaction process was intended from the beginning, as it is well known that compacting the monolith supports increasing the conductivity of the produced carbon monolith [17], it was aimed here to know the combined effect of pressing the green monolith and chemically activating it.

Figure 8 reveals that samples (A, B and C) are mesoporous materials with pore volume ratio (V_micro_/V_meso_ = 0.12). It is apparent that by lowering the concentration of the KOH solution, the distribution curve of mesopores shifts to higher adsorbed volume values.

Samples (D), (E) and (F) have the same micropore size distribution (Figure 9), except that the peak value for each distribution curve is slightly different. The micropore volume of sample (D) at the peak of the distribution curve is larger than that of sample (E), which in turn is larger than that of sample (F). This means that by decreasing the carbonization temperature, the micropore volume increases.

The mesopore size distributions of samples (A), (B) and (C) are almost the same (Figure 10), together they display the highest value of mesopore volume at the low values region of pore radius.

The mesopore size distributions of samples (D), (E) and (F) in Figure 11 are almost identical. Sample (D) exhibits the largest value of mesopore volume at small size mesopores. From Table 1, it is observed that the ratio of (V_micro_/V_meso_ = 3.1) is kept constant for the three samples, which means that the temperature of carbonization does not affect the proportion of micropore volume to mesopore volume.

#### 3.4.2. Electrical Impedance Spectroscopy (EIS)

The EIS data for samples (A), (B), and (C) are presented in Figure 12 as Nyquist plots over the frequency range 0.01–100,000 Hz. The curve representing sample (A) has a small half-semicircle at the high frequency region. This half-semicircle expresses the resistive nature of the electrode and the electrolyte combination [19]. The starting point of the semicircle represents the ohmic internal resistance (ESR) [7], where the equivalent series resistance (ESR) denotes the resistance to electrons and ions mobility in an electrochemical cell [20]. The two curves representing samples (B) and (C) did not show such semicircle segments, the value of ESR can be evaluated by the intercept of the curve line with the Z’ real axis [21].

The second segment in the curve representing sample (A), which is called the Warburg diffusion line (a straight line with a slope approximately 45 degrees), displays the capacitive and resistive characters of the ions penetrating into the pores of the electrode [19], it was noted that it has a short length which indicates that sample (A) may has capacitive characteristic [16]. For the curves representing samples (B) and (C), it was found that each of them have a long Warburg line with lower slope than that for sample (A), which means that samples (B) and (C) had probably lower capacitive characteristic than sample (A).

In general, there is a third segment in the Nyquist plot for supercapacitors, this third segment is a straight line that sharply increases at the low frequency region, and represents the dominance of the capacitive character of the electrode due to the formation of ionic and electronic charges of the electric double layer at the micropore surface of the electrode [19]. This straight line is absent in all curves of samples (A), (B), and (C), which indicates that these samples are lacking high supercapacitive behavior. Table 5 shows the values of ESR for samples (A), (B), and (C).

Furthermore, the EIS data for samples (D), (E), and (F) are presented in Figure 13 as Nyquist plots over the frequency range 0.01–100,000 Hz.

It is clear from Figure 13 that the three curves for samples (D, E, and F) do not have the third segment (nearly a vertical line) that represents the supercapacitive behavior. This indicates that samples (D, E, and F) do not exhibit high supercapacitive behavior [19]. An enlarged graph for Figure 13 at the high frequency region is given in the inset graph. This enlarged graph shows that sample (F) had a semicircle at the high frequency region. The diameter of this semicircle represents the charge transfer resistance (R_ct_) which denotes the resistance to the flow of ions and electrons at the electrolyte–electrode interface [22]. On the other hand, both samples (D and E) had semicircles with diameters less than sample (F), the values of the diameters of these semicircles (R_ct_) are shown in Table 5. The three samples (D), (E), and (F) have parallel Warburg lines with different lengths, indicating that they have different capacitive behavior. The value of ESR for each sample is shown in Table 5.

From Table 5 and Figure 13 and Figure 14 we conclude the following:Samples D, E, and F have better supercapacitive behavior than samples A, B, and C, because they displayed straight lines that lean more towards the vertical line in the high frequency region.Samples D, E, and F have lower ESR values than samples A, B, and C, which is considered better with regards to supercapacitor discharging power.Warburg lines for samples D, E, and F are significantly shorter than those of samples A, B, and C. This shows their greater capacitive behavior over A, B, and C samples.

#### 3.4.3. Cyclic Voltammetry 

It was observed that the cyclic voltammograms in Figure 14 for all the samples (A, B, C, D, E, and F) appeared without any peaks, this means that they exhibit electrical double–layer capacitance behavior and there are no redox reactions involved [23]. 

The CV test was performed in the potential range (−0.5 to 1 volt) for all the samples (A, B, C, D, E, and F). The symmetrical shape of all the voltammograms within that potential range indicates that these electrodes perform well without electrochemical reactions with the electrolyte.

The calculation of the specific capacity C_sp_ of the EDLC is based on the following equation [24]:(1)$$\mathrm{Csp}=\frac{1}{m\;∆V\;\nu }\int_{V1}^{V2}I\left(V\right)dV$$
where: $\int_{V1}^{V2}I\left(V\right)dV$ is the hysteresis loop area (VA)

ν is the voltage scan rate (V/sec)

∆V = V_2_ − V_1_ is the range of potential (V)

m is the weight of the active mass of the electrode (gram)

The hysteresis loop areas were calculated using ORIGIN software, the scan rate used for the calculation of the specific capacity for all the samples was 1 mv/sec. Results of calculation of C_sp_ for the different samples are shown in Table 6, along with their corresponding S_BET_.

It is observed from Table 6 that among the physically activated samples (D, E, and F), sample (F) has the highest specific capacity (29.25 F/g) compared to samples E and D. Furthermore, it is recognized that by increasing the temperature of activation of the physically activated samples, the specific capacity increases. For the chemically activated samples (A, B, and C), it is found that the specific capacity decreases when decreasing the specific surface area and increasing the concentration of the KOH solution. It is apparent here that physical activation of the prepared composite monolith is more appropriate than its chemical activation for obtaining higher values of specific capacity.

It is interesting here to give an interpretation for the result obtained in Table 5, as **S_BET_** increases for samples F, E, and D orderly, the corresponding **C_sp_** values decrease. The answer to this is that the BET analysis is done by the adsorption of a gas in the pores of the material, these pores may be too small to allow the entrance of ions and consequently the buildup of the stored charge within the electrode, therefore these small pores may be termed as blocked areas which may arise as a result of irregular long range pathways or collapsed channels within the electrode material [1]. If this interpretation is right, then it implies that using higher temperatures during physical activation provide a way to reclaim the irregular and collapsed pore channels. At the same time, it decreases the pore interspacing and the number of individual scattered pores, thereby decreasing the specific surface area of the electrode material and increasing its ion accessibility. 

It is desirable here to obtain cyclic voltammograms at different scan rates for sample F, as it represents the highest specific capacity obtained in this work. The cyclic voltammograms of sample (F) in Figure 15 display the goodness of reversibility and EDLC stability during charge and discharge. This advantageous performance of the EDLC is due to the storage of electric charges in the pores of the electrode material which are filled with the 2 M H_2_SO_4_ electrolyte during the charging process [17].

It is also observed that the electric current (I) corresponding to the largest voltage (1 volt) in the voltammograms of samples (D, E, and F) has the following order I_F_ > I_E_ > I_D_, while in the voltammograms of samples (A, B, and C), it has the following order I_B_ > I_A_ > I_C_. This result could be ascribed to the electrical resistivity or the internal porous structure of the electrode material which provides resistance to the flow of current. According to the previous surface area analysis, it was found that samples (D, E and F) have total pore volume 0.299, 0.266 and 0.249 cc/g, respectively. Surely, pore sites detract from the electrical conducting ability of the electrode, therefore as the pore volume decreases, it permits the passage of more current through the electrode. 

#### 3.4.4. Electrical Resistivity

The four-point technique was employed to measure the electrical resistivity of the produced electrode, it is shown in Figure 16. This technique involves contacting four equally-spaced, co-linear probes to the surface of the material under investigation. A DC voltage is applied between the outer probes (1 and 4), the resulting current (I) flowing between probes (4 and 1) and the voltage drop (V) between the two inner probes (2 and 3) are measured [25].

The sheet resistance (R_s_) is calculated from the following equation,
(2)$$Rs=\frac{\pi }{Ln\;2}*\frac{V}{I}$$

Then, the resistivity (ρ) of the material is given by:(3)ρ=Rs∗t
where t is the material sheet thickness.

Table 7 shows the values of the electrical resistivity of all the samples at 25 °C. It is seen from the table that the resistivity (ρ) of the physically activated samples follows the order ρ (F) > ρ (E) > ρ (D), this means that the resistivity increases by increasing the temperature of carbonization, where sample (D) has the lowest resistivity which in turn means that it has the highest conductivity (σ = 1/500 Ω.m = 0.002 S/m). It is also found from Figure 17 that the resistivity of samples (D) and (E) is kept constant from about −130 °C up to 80 °C, this represents good thermal stability of their electrode materials over that temperature range (measurement temperature range) which is larger than the temperature range (−40 °C to 65 °C) mentioned in reference [1]. This increased range of temperature where the electrochemical supercapacitor can operate is desirable as it extends its applicability over environments with different temperatures [1]. On the other hand, sample (F) shows slight increase in resistivity when increasing temperature from room temperature up to 80 °C. It is clear from the same figure that there is a steep increase in resistivity after 80 °C for all the samples.

For samples (A), (B), and (C) which were chemically activated, it is found from Figure 18 that samples (A) and (C) has nearly constant resistivity over the temperature range (25 °C to 80 °C), while there is a slight increase in resistivity for sample (B) over the same temperature range. Also, there is a gradual increase in resistivity after 100 °C for all the samples but with a lower change rate than those of samples (D), (E), and (F). Sample (B) has the least resistivity (highest conductivity 1/293 = 0.004 S/m) at room temperature. For comparison purposes, it is worth mentioning that activated carbon electrical conductivity is on the order of 10^−6^–10^2^ S/m [26]. One last feature which can be drawn from Figure 17 and Figure 18 is that there is a gradual decrease in resistivity for samples (A), (B), and (C) below 0 °C, although there is an approximate constancy in resistivity for samples (D), (E), and (F) below 0 °C. It should be noted here that pressing the green monolith during the curing process for samples (A, B, and C) reduced their electrical resistivity when compared to samples (D, E, and F) by decreasing the void’s volume (particle interspacing volume) within the electrode material. 

The electric resistivity values in Table 6 indicate that the resistivity (ρ) of samples (A, B, and C) follows the order (ρ_B_ < ρ_A_ < ρ_C_). This in turn explains the order (I_B_ > I_A_ > I_C_) in the voltammograms of Figure 10 (the higher the resistivity, the lower the current). 

#### 3.4.5. Galvanostatic Charge-Discharge

The galvanostatic charge–discharge (GCD) test was applied to all samples at constant specific current 50 mA/g in the voltage range (0–1) volts, supplied to the working electrode in the three electrode cell. 

For the physically activated samples (D, E, and F), it is observed from their GCD curves in Figure 19 that during the charging and discharging phases, the voltage varies linearly with time. At the other hand, it is found that the discharge curve is not equal in length to the charging curve for the three samples, where the electrode potential starts to appear at 0.5 volts for samples (F and D) and at 0.85 volts for sample E. This denotes that these samples have moderate capacitive behavior [17].

There is a sharp decrease of the potential at the beginning of the discharge curve for the three samples, this potential decrement ∆V is due to the equivalent series resistance ESR [16].

It is evident from Figure 19 that this voltage decrement follows the order ∆V_E_ > ∆V_D_ > ∆V_F_. This implies ESR_E_ > ESR_D_ > ESR_F_, which is consistent with the EIS Test.

The time duration of both charging and discharging of sample F is larger than those of samples E and D, which made the surface area under the GCD curve of sample F larger than that of samples E and D. Since the capacity level of the electrode is proportional to the area under the GCD curve [17], therefore the specific capacity of sample F is larger than the specific capacities of samples E and D. This is consistent with the previous cyclic voltammetry results.

For the chemically activated samples (A, B, and C), Figure 20 depicts their GCD plots. It is seen from these plots that both samples (A and C) is non-polarizable because their potential did not rise within the test range (0–1) volts, even when it rises, it does so abruptly with no time delay. This means that samples A and C are resistors and exhibit no capacitive behavior. There is an enlarged picture for sample A in Figure 20 to differentiate it from sample C. On the other hand, sample B shows some degree of capacitive characteristic as the electrode potential starts to rise gradually at 0.4 volts and then it discharges at 1 volt in small time duration. There is a sharp decrease in the voltage at the beginning of the discharge curve for sample B, this sharp decrease is associated with the ESR value. The ESR value was estimated here from the GCD curve, it was about 126 Ohm, much larger than the one estimated by the EIS test. 

Finally, from the GCD analysis of all the samples, it is found that samples (D, E, and F) are more favorable than samples (A, B, and C) to be utilized as supercapacitor electrode.

#### 3.4.6. Effect of Scan Rate on Specific Capacity

It is seen from Figure 21 for samples (A) and (B) that each curve follows three stages; the first stage has a large rate of decrease in the specific capacity when increasing the scan rate up to 10 mv/sec. Then in the second stage, there is a lower rate of decrease of the specific capacity till 25 mv/sec. While in the third stage, the lowest rate of decrease of specific capacity is seen from 25–100 mv/sec. Sample C shows no specific capacity.

The diminishing in the specific capacity which occurred with an increase of the scan rate could be interpreted as limited mass transfer kinetics of ions through the pores of carbon electrode, where there is a sluggish transport of ions through the electrode material with rapid changes of cell voltage [1].

For sample F, it is observed from Figure 22 that there are three stages of change in specific capacity with the scan rate, while there are two stages of decrease in the specific capacity with the scan rate for samples D and E. There is a large fall in the specific capacity of sample F from 100 F/g to about 30 F/g by changing the scan rate from 1 mV/sec to 10 mV/sec. There is a gradual decrease in the specific capacity of sample F when increasing the voltage scan rate within the range of 10–50 mV/sec. Then, a small change of the specific capacity when the scan rate is within the range of 50–100 mV/sec. For samples D and E, there is a slight change in the specific capacity when the scan rate is within the range of 1–10 mV/sec, and then there is a small change of the specific capacity when the scan rate is within the range of 10–100 mV/sec. The reason for this behavior of decrease in the specific capacity when increasing the scan rate for the three samples, D, E, and F, could be interpreted as before, due to the limited motion of ions through the electrode material.

#### 3.4.7. Capacity Retention

Figure 23 shows the capacity retention of the two samples F and B. These samples were chosen because they represent the optimum samples (with regards to the capacitive ability) of the physical activation and chemical activation procedures, respectively. It is seen from Figure 23 that sample F can retain about 91 % of its initial capacity at the 100th cycle, while sample B can retain about 95 % of its initial capacity.

#### 3.4.8. Ragone Plot

Ragone plot depicts the relationship between the specific power (P) and the specific Energy (E) [1,27], where:(4)$$E=0.5*\mathrm{Emax}*(1+\sqrt{1-\frac{P}{\mathrm{Pmax}})}$$
(5)$$\mathrm{Emax}=0.5*0.5{*\mathrm{Csp}*V}^{2}$$
(6)$$\mathrm{Pmax}=0.125*\frac{V^{2}}{\mathrm{ESR}*m}$$

E_max_ is the maximum specific energy in (Whr/kg) calculated for an electrode, C_sp_ is the specific capacity in (F/g), V is the potential window of the electrolyte, Pmax is the maximum specific power in (W/kg) calculated for an electrode, m is the active mass of the electrode and ESR is the equivalent series resistance.

Ragone plot for sample (F) will be given here as sample (F) represents the optimum electrode obtained in this work from both the capacitive behavior and the specific capacity value point of view. The following values are going to be used:

C_sp_ = 29.25 F/g, m = 1.3 g, ESR = 2.3 ohm, V = 1.5 volts

Upon inserting the previous values in Equations (4) and (5), the following is obtained:

E_max_ = 4.6 Whr/kg and P_max_ = 93 W/kg

The plot of Equation (4) is shown in Figure 24

For the sake of comparison of both E_max_ and P_max_ obtained here for sample (F) with those of composite monoliths cited in literature, Table 8 displays the E_max_, and the P_max_ values for some composite monolith electrodes such as activated carbon/graphene oxide, activated carbon/CNT and activated carbon/graphene.

In order to compare the results of this study with other previous studies about different types of supercapacitor carbon electrodes, we are listing in Table 9 the values of S_BET_, C_sp_, ESR of sample (F) obtained in this study besides those of other studies.

## 4. Conclusions

A supercapacitor electrode was prepared from widely available and cost effective raw materials, namely; textile-grade polyacrylonitrile fibers and phenolic resin. The procedure of preparation encompassed two different procedures of chemical activation and physical activation of the molded composite monolith. The main findings of this research work can be summarized as follows:

The produced composite electrode (by physical and chemical activation) had a mixture of micropores and mesopores. Applying a pressure on the molded monolith during curing (through the chemical activation procedure) resulted in obvious reduction in both the specific surface area and the electrical resistivity of the produced electrode (compared to the electrode obtained from the physical activation procedure). The highest conductivity obtained for the composite electrode was reached through chemical activation, it was 0.004 S/m approximately, compared to the conductivity of activated carbon electrodes in literature (10^−6^–10^2^ S/m).

Physical activation was more appropriate than chemical activation in providing a supercapacitive electrode, where the highest specific capacity obtained was 29.25 F/g in the potential window (−0.5–1 volt) compared to activated carbon and carbon Xerogel electrodes in literature with 233 F/g [30,33]. The electrode with that value showed good EDLC behavior and good capacitive behavior in regard to charging and discharging behavior, with almost constant resistivity over the temperature range −130 to 80 °C. The equivalent series resistance of the optimum electrode developed in this study was 2.3 ohm compared to 0.5 ohm, 0.36 ohm, and 101 ohm for activated carbon, templated carbon and carbon nanotubes electrodes developed in the literature, respectively. Furthermore, the charge transfer resistance of the optimum electrode was 0.33 ohm compared to 0.12 ohm, 0.41 ohm and 4.5 ohm for composite monoliths comprising (activated carbon, CNT and binder), (activated carbon, graphene and binder) and (activated carbon, carbon black and binder), respectively found in the literature [14,37,38]. Finally, the chemical activation procedure of the composite monolith prepared in this work resulted in poor capacitive electrode in regard to the charging and discharging behavior.

## Figures and Tables

**Figure 1 materials-13-00655-f001:**
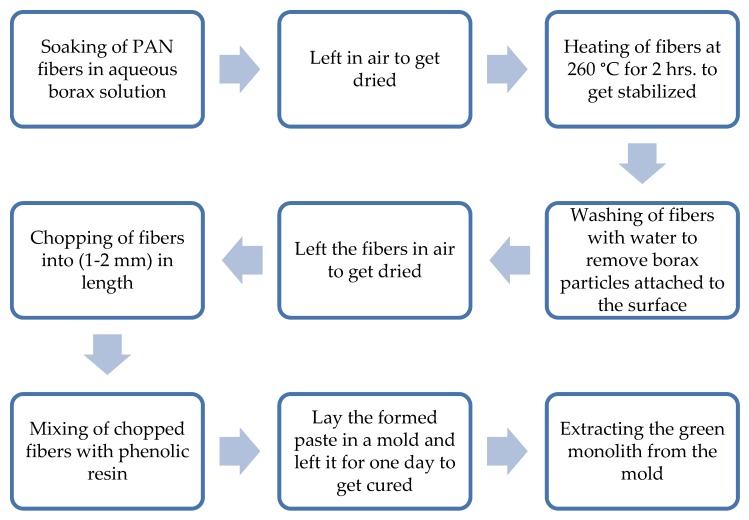
Steps of preparation of the composite green monolith. **Note:** The green monolith was laid under a pressure of 14 N/cm^2^ for eighteen hours at room temperature during the curing process for all the chemically activated samples, while the physically activated samples were not pressed. This was intended to observe the combination effect of pressing and chemical activation of the composite electrode.

**Figure 2 materials-13-00655-f002:**
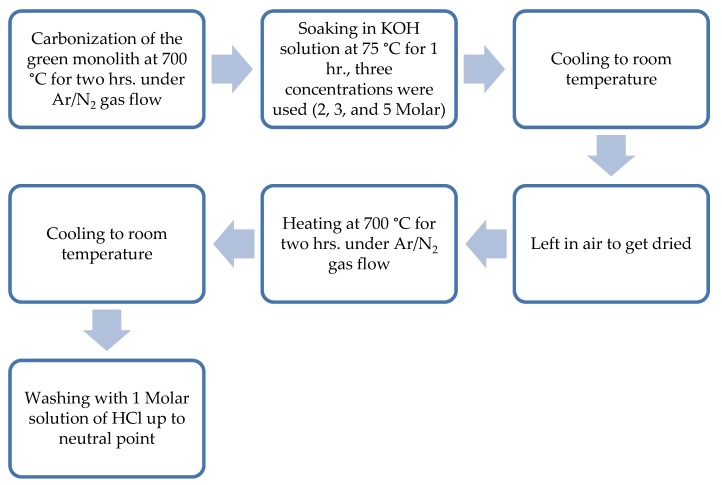
Steps of chemical activation of the composite green monolith.

**Figure 3 materials-13-00655-f003:**
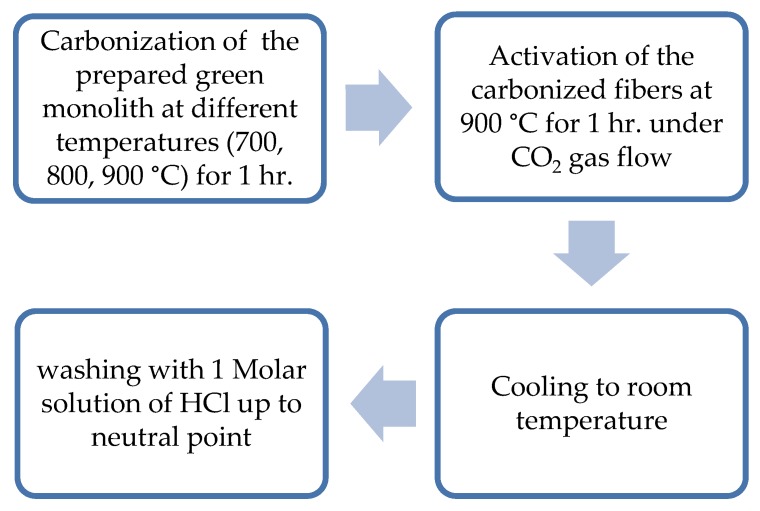
Steps of physical activation of the composite green monolith.

**Figure 4 materials-13-00655-f004:**
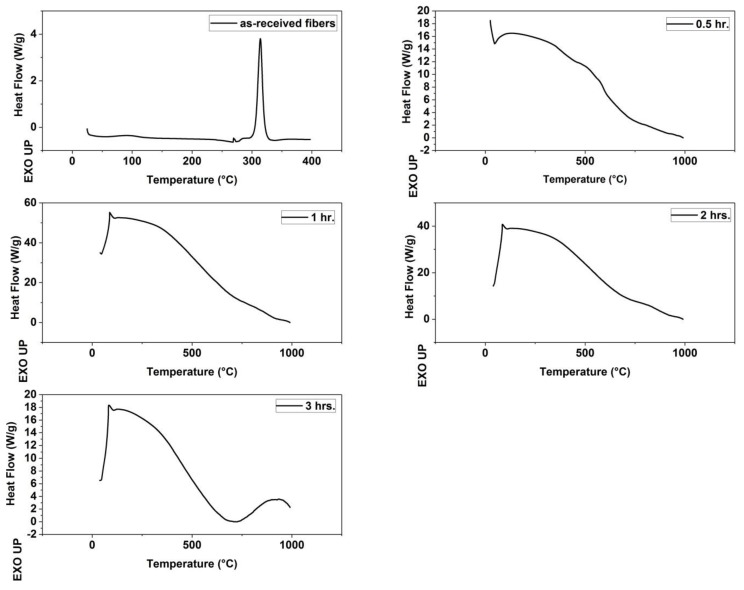
Differential scanning calorimetry (DSC) Thermograms of (as-received fibers, fibers stabilized for 0.5 h, fibers stabilized for 1 h, fibers stabilized for 2 h and fibers stabilized for 3 h).

**Figure 5 materials-13-00655-f005:**
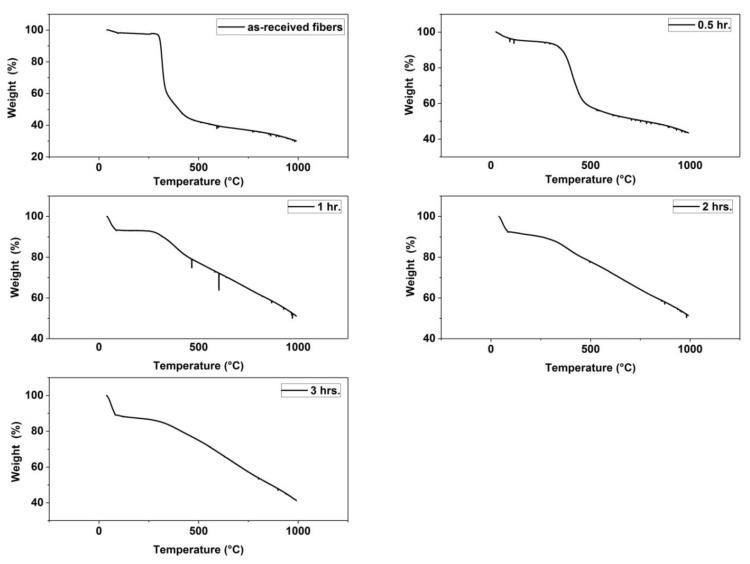
Thermal gravimetric analysis (TGA) Thermograms of (as-received fibers, fibers stabilized for 0.5 h, fibers stabilized for 1 h, fibers stabilized for 2 h and fibers stabilized for 3 h).

**Figure 6 materials-13-00655-f006:**
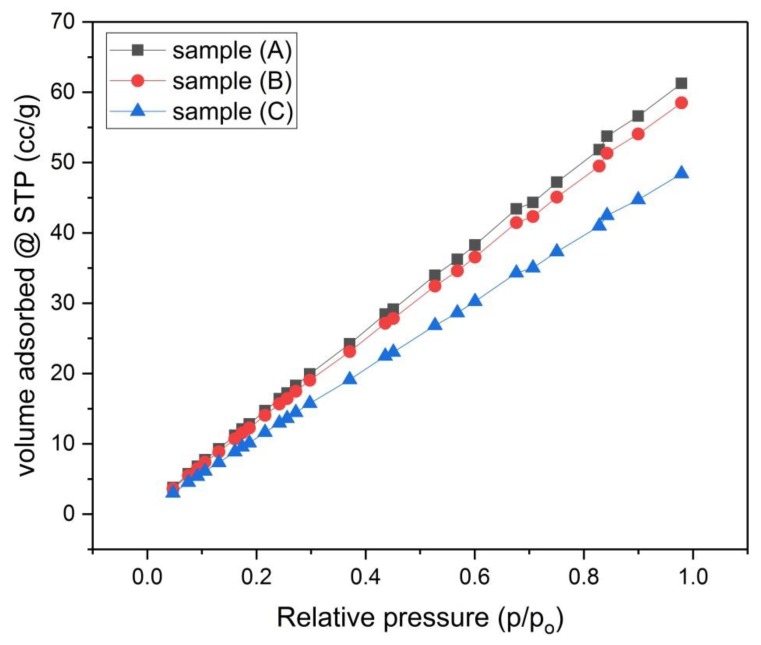
N_2_ Adsorption Isotherms for samples (A), (B) and (C).

**Figure 7 materials-13-00655-f007:**
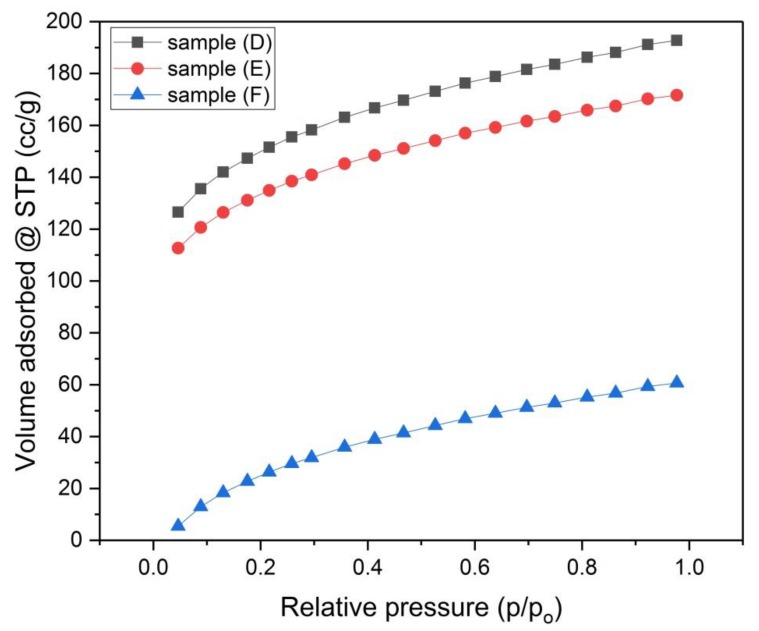
N_2_-Adsorption Isotherms for samples (D), (E) and (F).

**Figure 8 materials-13-00655-f008:**
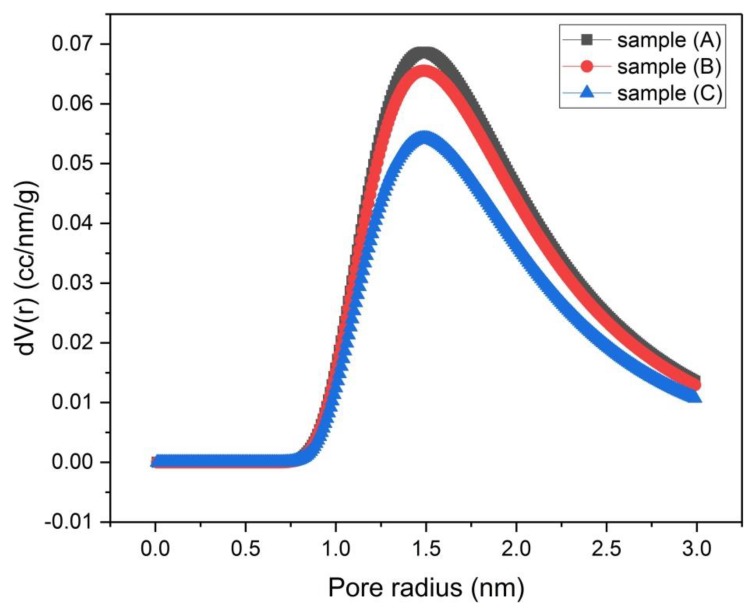
Micropore size distribution by the Dubinin–Astakhov (DA) method for samples (A), (B) and (C).

**Figure 9 materials-13-00655-f009:**
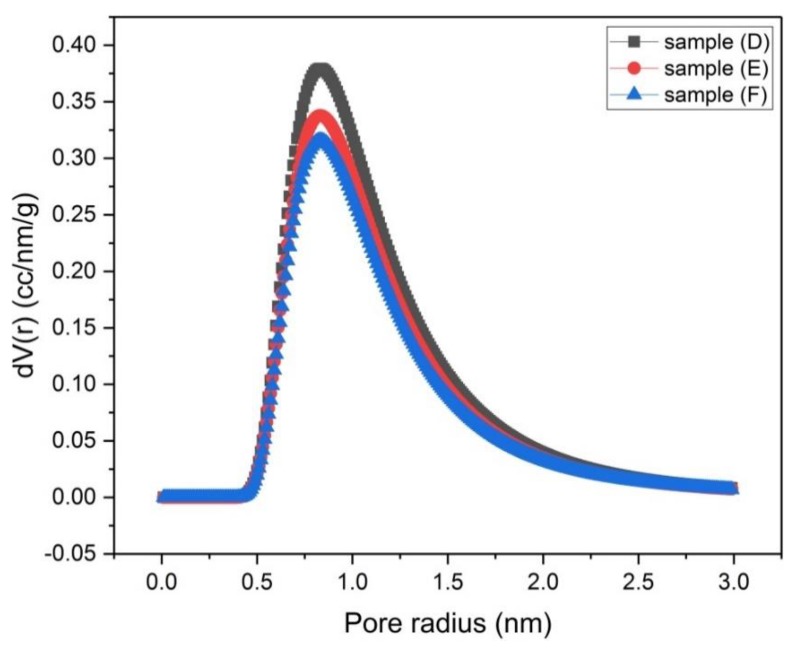
Micropore size distribution by the DA method for samples (D), (E) and (F).

**Figure 10 materials-13-00655-f010:**
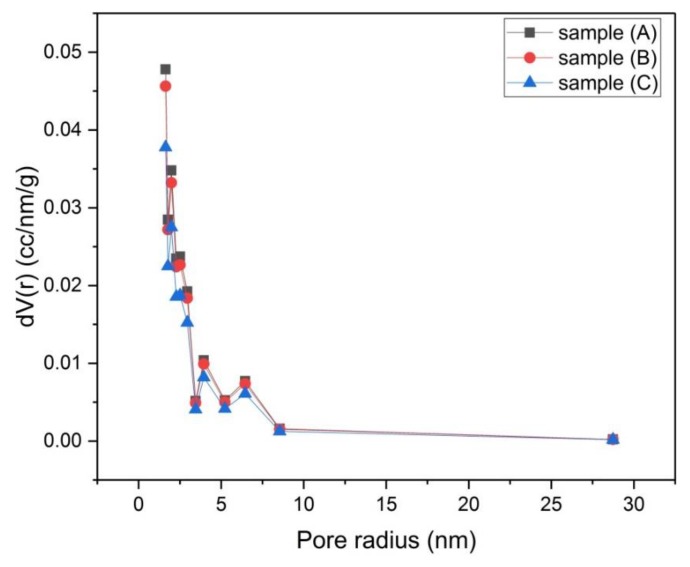
Mesopore size distribution by the Barrett–Joyner–Halenda (BJH) method for samples (A), (B) and (C).

**Figure 11 materials-13-00655-f011:**
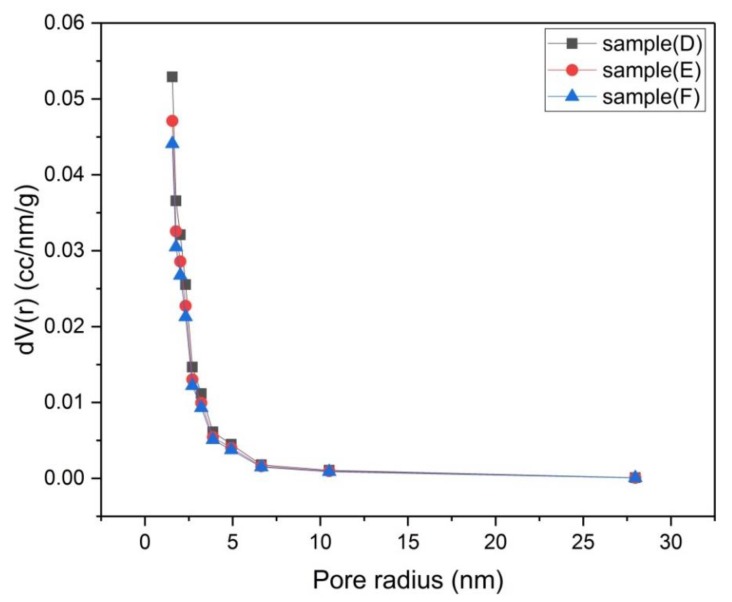
Mesopore size distribution by BJH method for samples (D), (E) and (F).

**Figure 12 materials-13-00655-f012:**
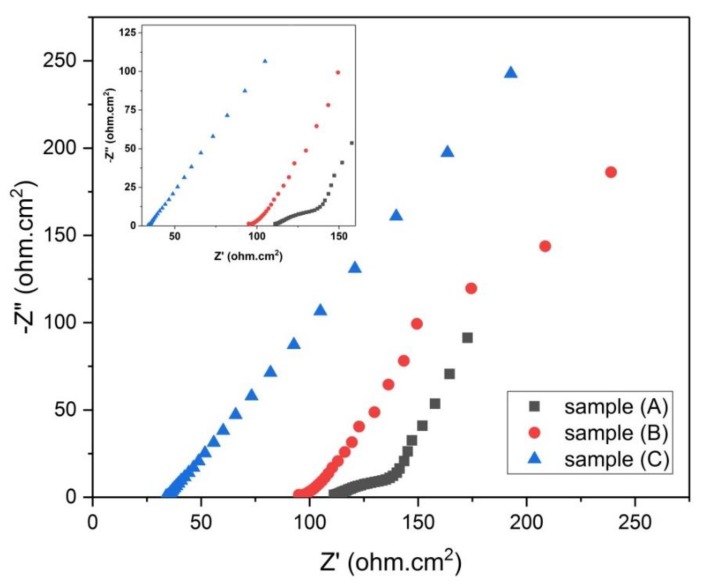
Nyquist plot for samples (A, B and C).

**Figure 13 materials-13-00655-f013:**
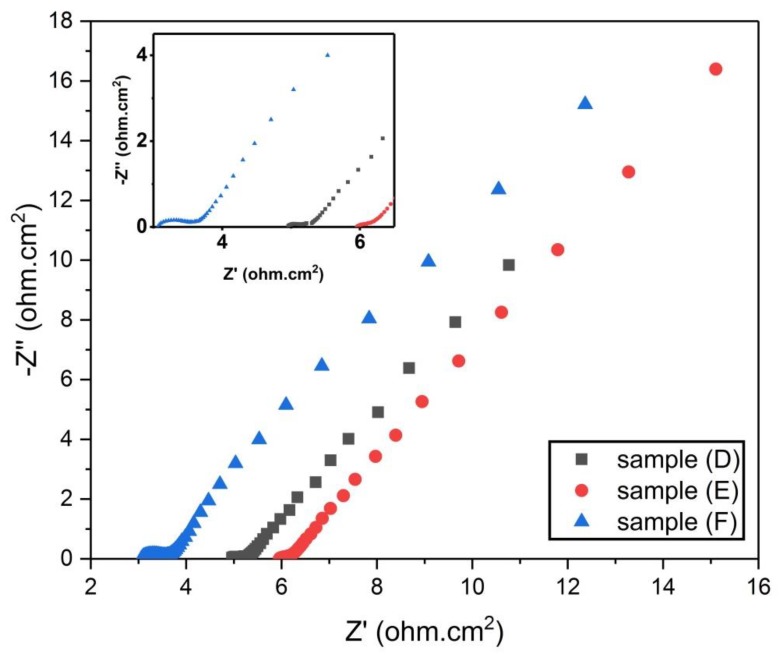
Nyquist plot for samples (D, E, and F).

**Figure 14 materials-13-00655-f014:**
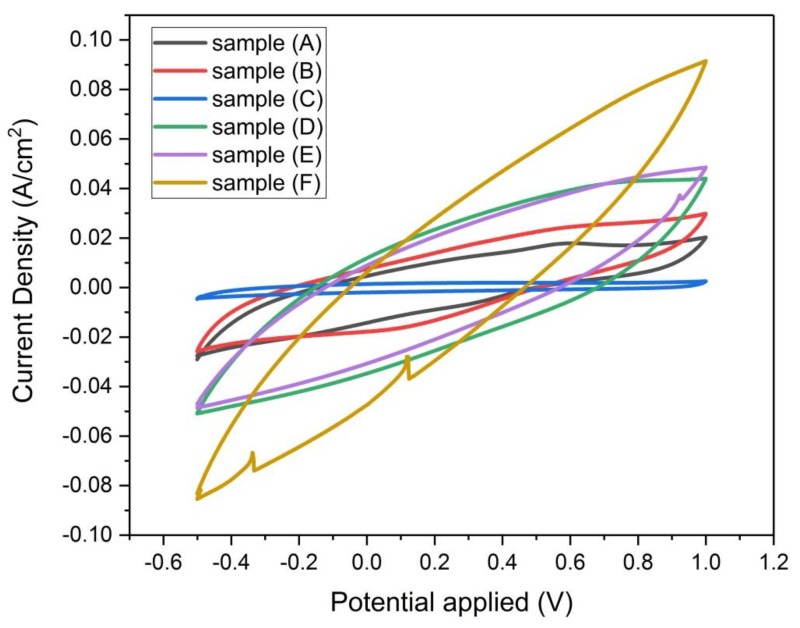
Cyclic Voltammograms of samples (A, B, C, D, E, and F) at scan rate 100 mv/sec, based on three-electrode measurement with 2 Molar H_2_SO_4_ solution.

**Figure 15 materials-13-00655-f015:**
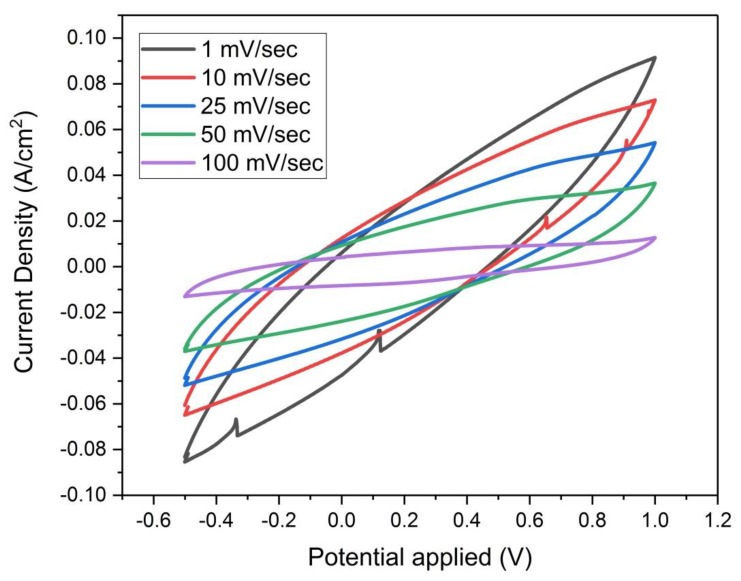
Cyclic Voltammograms of sample (F) at scan rates (1, 10, 25, 50 and 100 mV/sec), based on three-electrode measurement with 2 Molar H_2_SO_4_ solution.

**Figure 16 materials-13-00655-f016:**
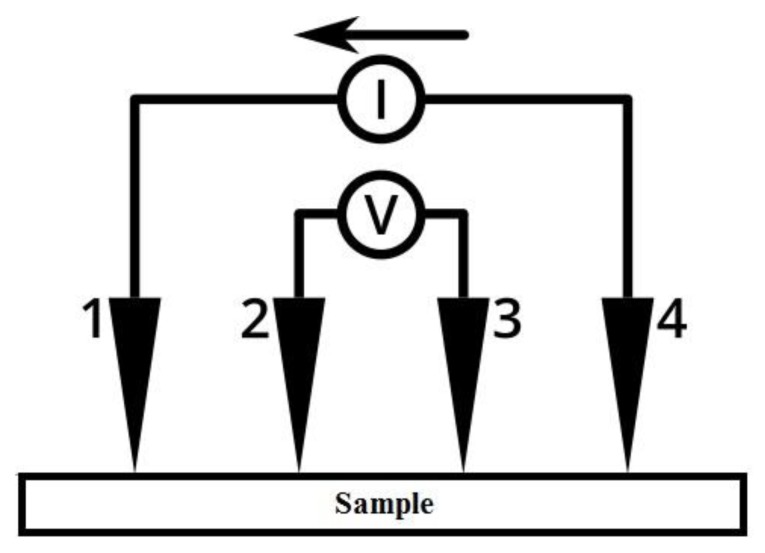
Four-point probe measurement principle.

**Figure 17 materials-13-00655-f017:**
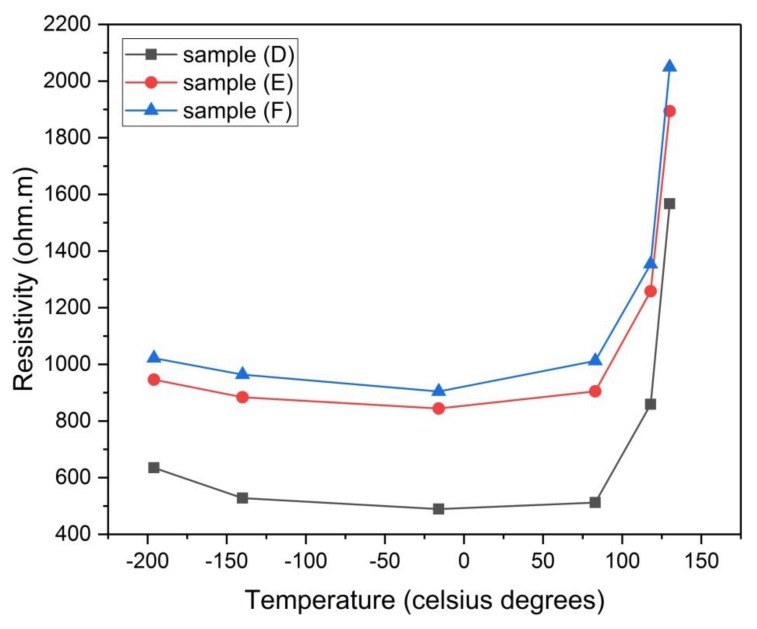
Electrical resistivity of samples (D), (E) and (F) versus the temperature of analysis.

**Figure 18 materials-13-00655-f018:**
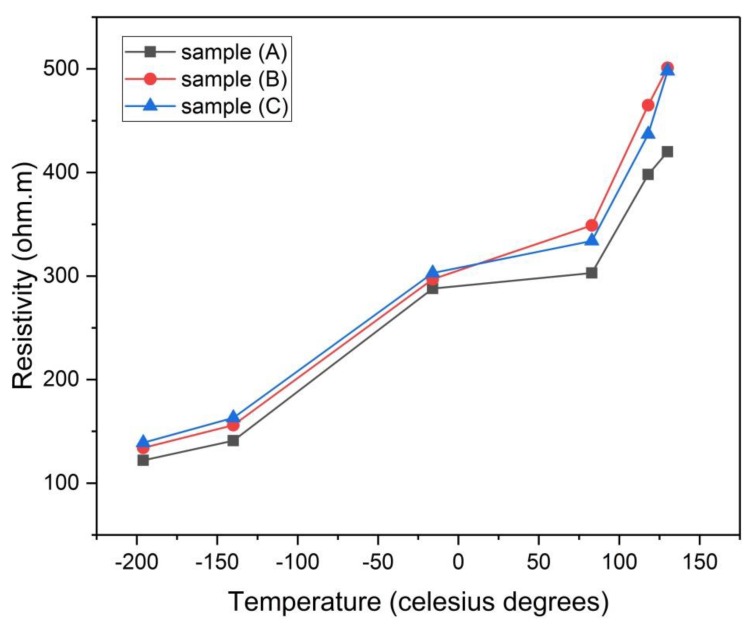
Electrical resistivity of samples (A), (B), and (C) versus the temperature of analysis.

**Figure 19 materials-13-00655-f019:**
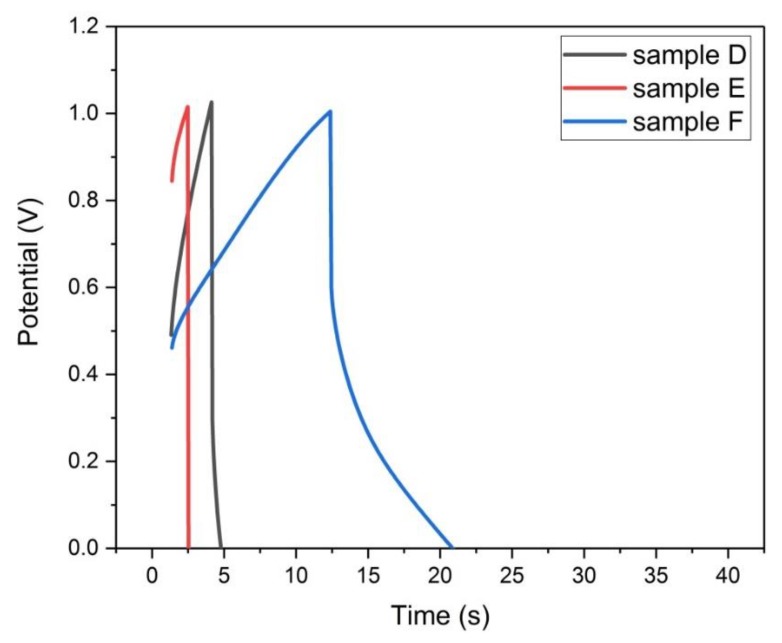
Galvanostatic charge–discharge plots of samples (D, E, and F) at 50 mA/g.

**Figure 20 materials-13-00655-f020:**
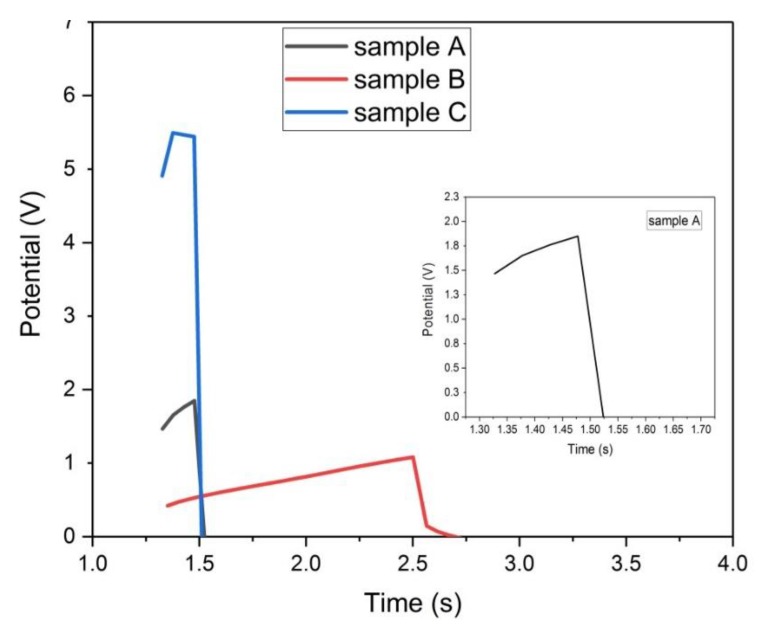
Galvanostatic charge-discharge plots of samples (A, B, and C) at 50 mA/g.

**Figure 21 materials-13-00655-f021:**
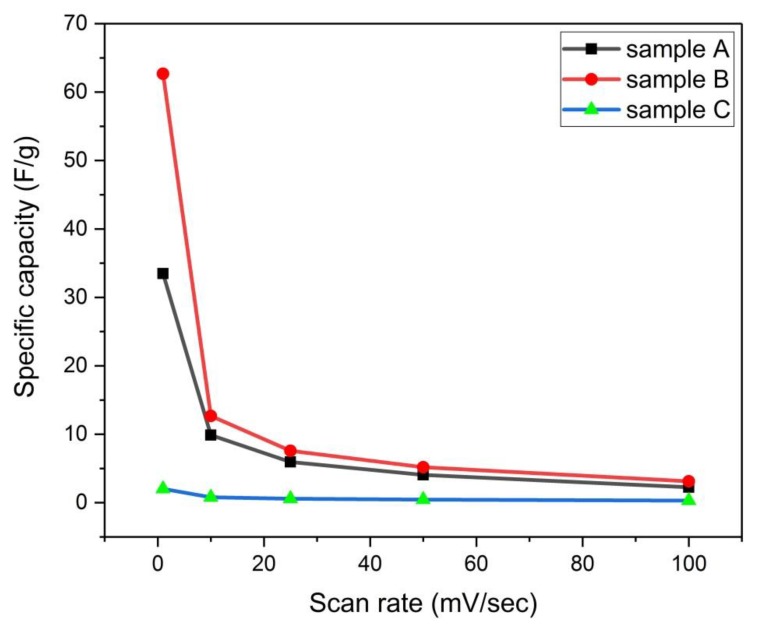
Variation of specific capacity with scan rate for samples (A, B, and C).

**Figure 22 materials-13-00655-f022:**
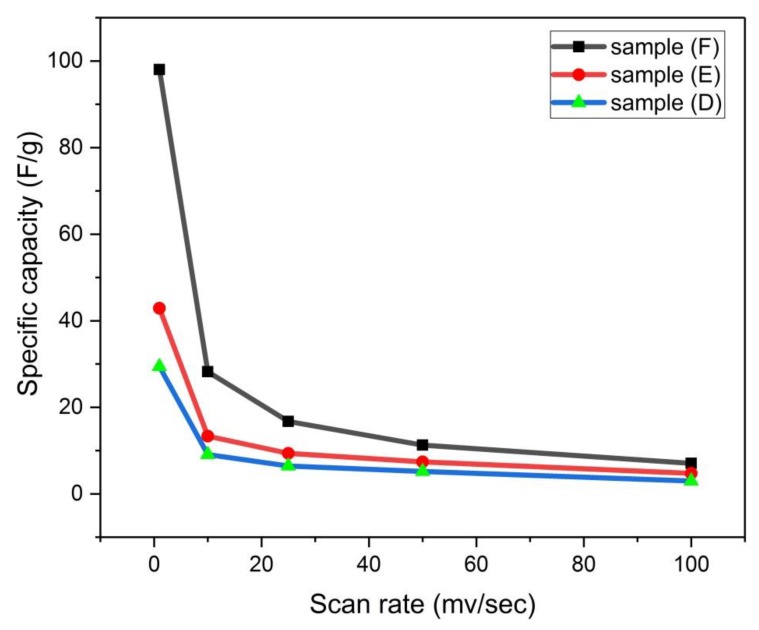
Variation of specific capacity with scan rate for samples (D, E, and F).

**Figure 23 materials-13-00655-f023:**
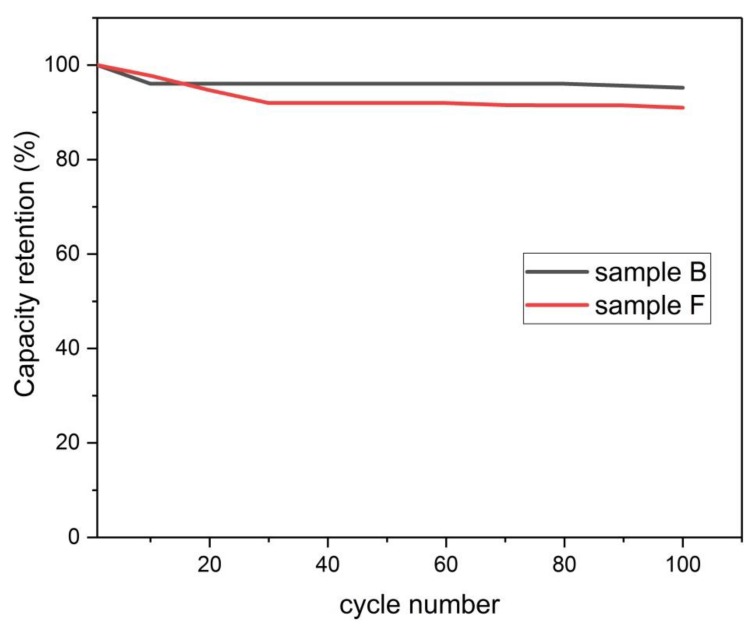
Capacity retention versus cycle number.

**Figure 24 materials-13-00655-f024:**
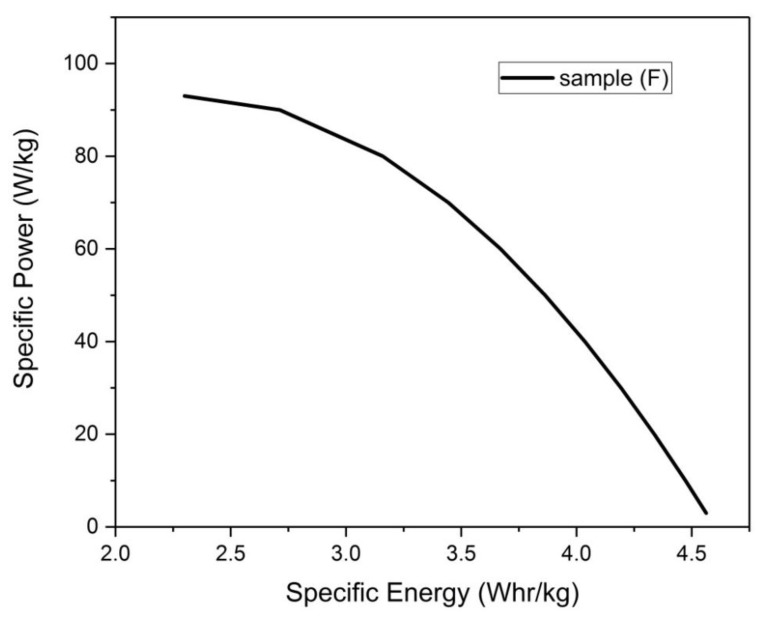
Ragone plot for sample (F).

**Table 1 materials-13-00655-t001:** DSC data of as-received fibers.

Type of Fibers	Energy Released (J/g)	Temperature at the Start of the Exothermic Peak (°C)	Temperature at the Vertex of the Exothermic Peak (°C)
As-received Fibers	251.3	305.91	314.06

**Table 2 materials-13-00655-t002:** Residual weights of the as-received fibers and thermally stabilized fibers.

Sample	Residual Weight (%)
As-received fibers	30.125
Fibers resided for 0.5 h in furnace	43.56
Fibers resided for 1 h in furnace	51.1
Fibers resided for 2 h in furnace	51.4
Fibers resided for 3 h in furnace	41.6

**Table 3 materials-13-00655-t003:** Elemental Analysis of as-received fibers and thermally stabilized fibers.

Sample	C %	H %	N %	C/H	C/N
As-received fibers	66.55	5.43	16.27	12.25	4.1
Fibers treated for 0.5h	71.28	6.14	18.99	11.6	3.75
Fibers treated for 1 h	70.27	3.2	26.46	21.95	2.65
Fibers treated for 2 h	57.34	1.4	20.25	40.96	2.83
Fibers treated for 3 h	65.12	4.3	23.09	15.14	2.82

**Table 4 materials-13-00655-t004:** Porous Structure Characteristics.

Sample	S_BET_ (m^2^/g)	S_micro_ (m^2^/g)	S_meso_ (m^2^/g)	V_micro_ (cc/g)	V_meso_ (cc/g)	V_total_ (cc/g)	V_micro_/V_meso_
(A)	100	42.76	57.24	0.0053	0.085	0.0903	0.062
(B)	95	41	54	0.0095	0.0812	0.0907	0.12
(C)	79	34	45	0.008	0.067	0.075	0.12
(D)	486	435	51	0.226	0.073	0.299	3.1
(E)	433	387	46	0.202	0.064	0.266	3.2
(F)	405	362	43	0.189	0.060	0.249	3.2

Note: V_Total_ = V_micro_ + V_meso_.

**Table 5 materials-13-00655-t005:** The ESR and R_ct_ values for different samples.

Sample	ESR (ohm/cm^2^)	R_ct_ (ohm/cm^2^)
A	33.88	-
B	29	-
C	11.74	-
D	1.36	0.049
E	1.18	0.028
F	1.73	0.248

**Table 6 materials-13-00655-t006:** C_sp_ for Different Samples and their Corresponding S_BET._

Sample	C_sp_ (F/g)	S_BET_ (m^2^/g)
A	6.0	100
B	5.12	95
C	0.78	79
D	7.28	486
E	9.94	433
F	29.25	405

**Table 7 materials-13-00655-t007:** Electrical Resistivity of Different Samples at 25 °C.

Sample	Resistivity (Ω.m)
A	294
B	293
C	303
D	500
E	932
F	1035

**Table 8 materials-13-00655-t008:** E_max_ and P_max_ values of some composite monolith electrodes.

Electrode	E_max_ (Whr/kg)	P_max_ (W/kg)	Ref.
Textile grade PAN fibers/phenolic resin	4.6	93	Present study
activated carbon/graphene oxide	6	30	[20]
activated carbon/CNT	11.25	3650	[28]
activated carbon/graphene.	8.5	25	[29]

**Table 9 materials-13-00655-t009:** S_BET_, C_sp_ and ESR of various carbon electrodes.

Type of Carbon Electrode	S_Bet_ (m^2^/g)	C_sp_ (F/g)	ESR (Ω)	Reference
Activated carbon monolith	405	29.25	2.3	present study
Carbon nanotubes (CNT)	430–1600	180	101	[30,31]
Graphene sheets	2630	150	96	[30,32]
Activated carbon	300–2749	100–233	0.5	[32,33]
Carbide derived carbon	1822	134	0.39	[34]
Templated carbon	1120	132	0.36–0.8	[35]
Carbon Xerogel	1243	234	33	[36]

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
