# Peer review of "Preparation of Composite Monolith Supercapacitor Electrode Made from Textile-Grade Polyacrylonitrile Fibers and Phenolic Resin"

_materials, 2020, doi:10.3390/ma13030655_

Round 1

Reviewer 1 Report

In the last sentence of the abstract the authors mention that chemical activation results in a very poor electrode. However, the authors do not state any performance metrics to prove that the electrode undergoing chemical activation acts more resistive. How are electrochemical double layer capacitors abbreviated as EDLSCs ? On Line 51, replace "continue up till now" with "are still in progress". On Line 57, delete "amorphous". On Line 82, delete "which will in turn lower its price in the market". On Line 93, replace "experimental work involved the" with "schematic for". On Line 128, replace "designated by" with "termed as". On Line 143, replace "were tested under nitrogen gas" with "were dried in nitrogen gas at ambient conditions" On Line 159, replace Impedance Spectroscopy with "Electrical Impedance Spectroscopy" On Line 161, replace "and to assure the capacitive behavior of the electrode in charging and discharging" with "and to evaluate the electrochemical performance of the electrode". Change deg. c to capital letters throughout the manuscript. On Line 201, delete the sentence "Firstly, the fibers neither degraded.... temperatures". On Line 209, delete "and their final weights.... respectively". On Line 228, explain in what context the amount of carbon in the electrode is important. On Line 229, replace "gradually for the fibers resided....." with "gradually with rise in treatment time from 0.5 to 2 hrs and then increases suddenly at 3 hrs." On Line 241, since the samples prepared using procedure 3 were damaged, please remove procedure 3 entirely from the paper. On Line 246, please include a brief paragraph on the importance of porosity parameter. Do electrodes with high porosity always exhibit high electrochemical surface area ? Please explain with references. On Line 259, explain what is type IV pattern. In Table 5, report ESR and Rct with area normalized resistance values. In the caption of Figure 10, mention the electrolyte and that it is a three-electrode measurement. On Line 376, define which weight is used to normalize capacitance. Rephrase the sentence in Line 382: " that there is a general trend ... the specific surface area" On Line 418, explain the test setup used to perform these measurements. In Figure 18, only the capacitance retention % of samples B and F are shown. What about other samples ? In the conclusion section, include a summary of the electrochemical performane of the electrode developed and comparison with past studies. Feel free to use Ragone plot to make your point.

Reviewer 2 Report

To Jessie Tong
Section Managing Editor, MDPI

Dear Editor,

Here is the review with comments for the manuscript (ID_materials-681920)

TITLE: Preparation of Composite Monolith Supercapacitor Electrode made from
Textile-grade Fibers and Phenolic Resin

In this paper, the author shared a supercapacitor composite material by textile grade PAN and Phenolic resin. Through activation, they measure key properties of the electrode materials such as specific capacitance, temperature range for performance, and etc. overall, I see there needs to have some big improvement for data analysis for their conclusion. Please modify for a revision.

For Fig. 13, the EIS curve looks unprofessional. Please check how many data points being collected. 14, you could enlarge the high and mid frequency part of the plot for analysis. 10, for capacitor, specific current based on footpint area will be a more reasonable way. You could also refer to Y. Gogoitsi’s paper. Table 6, through put electric resistivity high depends on experimental setup. Eg. pressure control and etc. Please have more details of that. 14, same as EIS plot, the data points is not enough to judge especially for sample E and D.

Round 2

Reviewer 2 Report

To Jessie Tong
Section Managing Editor, MDPI

Dear Editor,

Here is the review with comments for the manuscript (ID_materials-681920-V2)

TITLE: Preparation of Composite Monolith Supercapacitor Electrode made from
Textile-grade Fibers and Phenolic Resin

In this paper, the author shared a supercapacitor composite material by textile grade PAN and Phenolic resin. Through activation, they measure key properties of the electrode materials such as specific capacitance, temperature range for performance, and etc. In this revised version, the quality has been improved greatly. However, I still suggest to improve some wording. Following are some points need to be address.

Overall, it is okay for publication without further review.

Thank you

Line 157, electrochemical impedance (as both happening) Line 229 to 231 should be revised for clarity, even though I can understand by reading several times. Fig 12, the curve for A, it is highly likely due to porous structure or tortuosity of the structure for the initial linear part (High to mid frequency)

This manuscript is a resubmission of an earlier submission. The following is a list of the peer review reports and author responses from that submission.

Round 1

Reviewer 1 Report

The general comments are below. Detailed comments can be found in the pdf attached.

The paper discusses the methods of preparation in much greater detail than required. Many of the details can be moved to the supplementary section. The figures are not labelled in detail. The x-axis and y-axis in each graph should be explained clearly in the labels. The GCD measurements shown in Figure 14 and 15 were performed only at a single value of operating current.  Literature survey is inadequate. References should be made to previous works on carbon-based electrodes and suitable comparisons in performance should be made. Please cite: ACS Appl. Energy Mater.20181115800-5809, Electrochimica Acta, Vol. 281, 2018, pp.357-369. Conclusions should focus on major findings rather than repeating processing techniques.

Reviewer 2 Report

The manuscript entitled” Preparation of Composite Monolith Supercapacitor Electrode made from Textile-grade Polyacrylonitrile Fibers and Phenolic Resin” describes the preparation of supercapacitor electrode from the cost-effective raw materials, textile-grade polyacrylonitrile fibers and phenolic resin. This electrode showed interesting chemical and physical activation properties. This will be an interesting manuscript if the author addresses below mentioned comments.

Why the capacitance is very low for highly porous materials?

What is the capacitance as a function of cycle number at constant current charge-discharge?

Authors describe the novelty of the work in the introduction.

It is useful to put a schematic rather than describing the procedure.

Does DSC data have baseline issues?

It is useful to run impedance on the cell to describe the different contributions like bulk resistance, charge transfer processes